# Interleukin-6 and Its Soluble Receptor Complex in Intensive Care Unit COVID-19 Patients: An Analysis of Second Wave Patients

**DOI:** 10.3390/pathogens12101264

**Published:** 2023-10-20

**Authors:** Gaetano Di Spigna, Daniela Spalletti Cernia, Bianca Covelli, Maria Vargas, Valentina Rubino, Carmine Iacovazzo, Filomena Napolitano, Loredana Postiglione

**Affiliations:** 1Department of Translational Medical Sciences, University of Naples “Federico II”, 80131 Naples, Italy; gaetano.dispigna@unina.it (G.D.S.); danielaspalletticern@libero.it (D.S.C.); valentina.rubino@unina.it (V.R.); filomena-napolitano88@hotmail.it (F.N.); loredana.postiglione@unina.it (L.P.); 2Department of Neurosciences, Reproductive and Odontostomatological Sciences, University of Naples “Federico II”, 80131 Naples, Italy; vargas.maria82@gmail.com (M.V.); iacovazzo@tin.it (C.I.); 3Center for Basic and Clinical Immunology Research (CISI), University of Naples “Federico II”, 80131 Naples, Italy

**Keywords:** acute respiratory distress, COVID-19, cytokine storm, IL-6 and soluble receptor complex, comorbidity index, second wave of infection

## Abstract

In December 2019, a SARS-CoV-2 virus, coined Coronavirus Disease 2019 (COVID-19), discovered in Wuhan, China, affected the global population, causing more than a million and a half deaths. Since then, many studies have shown that the hyperinflammatory response of the most severely affected patients was primarily related to a higher concentration of the pro-inflammatory cytokine interleukin-6, which directly correlated with disease severity and high mortality. Our study analyzes IL-6 and its soluble receptor complex (sIL-6R and sgp130) in critically ill COVID-19 patients who suffered severe respiratory failure from the perspective of the second COVID wave of 2020. A chemiluminescent immunoassay was performed for the determination of IL6 in serum together with an enzyme-linked immunosorbent assay to detect serum levels of sIL-6R and sgp130, which confirmed that the second wave’s serum levels of IL-6 were significantly elevated in the more severe patients, as with the first 2019 COVID-19 wave, resulting in adverse clinical outcomes. At present, considering that no specific treatment for severe COVID-19 cases in its later stages exists, these molecules could be considered promising markers for disease progression, illness severity, and risk of mortality.

## 1. Introduction

An unexpected pandemic, caused by a SARS-CoV-2 virus known as Coronavirus Disease 2019 (COVID-19) identified in Wuhan, China, broke out in December 2019, affecting the global population and causing more than 77 million infections and over one and a half million deaths [1,2,3], making it urgent to develop an anti-COVID-19 vaccine or identify an anti-COVID-19 therapeutic drug to counteract the rapidly spreading virus [4,5]. SARS-CoV-2 is a positive-sense, single-stranded RNA (+ssRNA, single linear RNA molecule) virus and shares major structural and molecular characteristics with other coronaviruses, including the presence of structural proteins S (spike), E (envelope), and M (membrane), responsible for the formation and stability of the viral envelope and N (nucleocapsid), which interacts with the RNA genome [6,7].

Many studies have focused on the virus’ structure and life cycle and its similarities with other coronaviruses, along with its ability to adapt to external pressure and factors affecting pathogenicity and comorbidity influence. However, less is known about the SARS-CoV-2 pathogenesis, although it is clear that a close interaction between the virus and the immune system results in different clinical manifestations of the disease [8]. While the majority of infected people are asymptomatic or have a mild case, 14% have been found to have developed a severe form, and 5% became critically ill with major complications, such as interstitial pneumonia, respiratory failure with acute respiratory distress syndrome (ARDS) [9,10,11], and multiple organ dysfunction syndromes (MODS) [12,13], considering that the virus infects the cells in the lower respiratory system, inducing a rapid local immune response.

Many experimental studies and clinical trials suggest that the overproduction of cytokines, known as a cytokine storm, is the consequence of an aberrant immune response and directly correlates with tissue damage and an unfavorable prognosis of severe lung diseases [14]. These pleiotropic cytokines are produced during sepsis, resulting in acute organ injury instigated by a variety of different cell types, including macrophages, lymphocytes, endothelial cells, epithelial cells, and fibroblasts, which are released into circulation at sites of tissue inflammation [15]. Severely affected patients, compared to those who are only moderately ill, have been found to have a hyperinflammatory response due to the higher concentration of the pro-inflammatory cytokine inter-leukin-6 (IL-6) [16,17,18,19,20], which leads to a decidedly poor prognosis [21,22]. Indeed, different therapies, such as tocilizumab, sarilumab, and steroids, are now allowed to modulate the hyperinflammatory activation in COVID-19 infection.

IL-6 is amongst the most important in the cytokine network and plays a central role in acute inflammation, human metabolism, and autoimmune cell differentiation [23,24,25] and is significant in the evolution of sepsis, especially as an early indicator of the inflammatory state. Several studies have detected nucleotide changes in the gene that codes for IL-6, which generates polymorphisms that may be related to risk factors or protectors to developing sepsis, septic shock, and even death [26,27]. IL-6 can be produced by almost all stromal and immune system cells, including B-lymphocytes, T-lymphocytes, macrophages, monocytes, dendritic cells, mast cells, and other non-lymphocytic cells such as fibroblasts, endothelial cells, keratinocytes, glomerular mesangial cells, and tumor cells [28] and therefore plays a key role in a cytokine storm; however, it is not the only molecule involved. We therefore chose to focus our study on not only on the role of IL-6 in COVID-19 disease severity but also on its receptors, membrane receptors gp130, and its soluble antagonist form (sgp130), including the soluble receptor agonist (sIL-6R) and its regulation through the aforementioned receptors [29,30].

In a classic pathway, IL-6 binds to IL-6-transmembrane-non-signaling receptor (IL-6R) to form a complex that subsequently binds to the transmembrane glycoprotein gp130, inducing its homo-dimerization and initiating intracellular signal transduction [31] occurring in cells that express IL-6R. IL-6R also exists in a soluble form (sIL-6R) and is released from cell surfaces by proteolysis and splicing of IL-6R mRNA. IL-6 can bind to this soluble molecule, thereby increasing the circulating IL-6 half life and forming a complex that interacts with gp130 to trigger downstream trans-signal transduction and gene expression. To this extent, sIL-6R is considered an agonist molecule [32,33], as it allows IL-6 to act on cells lacking IL-6R yet expressing the ubiquitous gp130. In addition, soluble forms of gp130 (sgp130) are released by cells as a consequence of its mRNA splicing. Although sgp130 cannot bind IL-6 or IL-6R alone, it can bind the IL-6/IL-6R complex, thus behaving as an antagonist molecule that can reduce or block the trans signaling mechanism [34].

The aim of our study was the evaluation of IL-6 and its soluble receptor complex in critically ill COVID-19 patients with Acute Respiratory Distress Syndrome during the second, 2020 wave of infection to evaluate a possible mechanism of disease progression as correlated to different comorbidity factors. We focused both on the role of IL-6 in disease severity as well as its receptors, membrane receptors gp130 and soluble antagonist form (sgp130), including the soluble receptor agonist (sIL-6R) and its regulation via these receptors.

## 2. Patients and Methods

### 2.1. Patients

A prospective observational study was carried out in the intensive care unit (ICU) of the University of Naples “Federico II” hospital during the second surge of the COVID-19 pandemic from September to December 2020 [35,36]; at that time, the COVID-19 vaccine was not available in Italy. The vaccination for COVID-19 was only made available for administration in January–February 2021.

The Ethics Committee (Azienda Ospedaliera Universitaria “Federico II”-Naples, protocol number: 155/120) approved the investigative protocol, and written informed consent was obtained from each patient or next of kin. All human study procedures were performed in accordance with the principles outlined by the Declaration of Helsinki. All adult patients with laboratory-confirmed COVID-19 infection (N = 104) admitted to our ICU for severe respiratory failure were included. A confirmed case of COVID-19 was defined by a positive result on a reverse transcriptase polymerase chain reaction (RT-PCR) assay of a specimen collected on a nasopharyngeal swab. A comprehensive data collection was designed to include comorbidities, clinical and biochemical characteristics, therapies, and outcomes of the COVID-19-infected patients. Patients were classified in “survivor” and “non survivor” groups according to their exitus. A Charlson comorbidity index was calculated for all patients assessing their levels of comorbidity and taking into account both the number and severity of 19 pre-defined comorbid conditions, which provided us with a weighted score, which was then used to predict short-term and long-term outcomes, including function, hospital length of stay, and mortality rates.

### 2.2. Methods

A chemiluminescent Immunoassay (CLIA) was performed for IL-6 serum determination by IMMULITE 2000 (SIEMENS Healthcare Diagnostics, Milan, Italy). Serum levels of sIL-6R and sgp130 were determined using an automated enzyme-linked immunosorbent assay (ELISA) by Triturus System (GRIFOLS, Naples, Italy). R&D Quantikine ELISA Kits (R&D Systems, Minneapolis, MN, USA—DIACHEM s.r.l., Naples, Italy) were employed for all determinations. The intra-assay and inter-assay variation coefficients were ˂5% for IL-6 and sgp130 serum levels, and ˂10% for sIL-6R, respectively.

Healthy control subjects (N = 98) without a family history of COVID-19 and with a mean age of 62.13 ± 9.28, recruited from employees of the Azienda Ospedaliera Universitaria “Federico II”, Naples, were enrolled for reference values evaluations and patient comparisons. Patient blood samples were taken at 3 intervals every 48 h to determine levels of IL-6, sIL-6R, and sgp130. Intra-assay and inter-assay variation coefficients were ˂5% for IL-6 and sgp130 serum levels and ˂10% for sIL-6R, with detection limits of: IL-6 (0.01 ÷ 350.24 pg/mL), sIL-6R (3.01 ÷ 232.82 ng/mL), and sgp130 (0.01 ÷ 2.40 ng/mL). 

### 2.3. Statistical Analysis

Data were expressed as mean ± standard deviation (S.D.). Data statistical evaluation by InStat 3.0 software (GraphPad Software Inc., San Diego, CA, USA) was carried out by a Mann–Whitney test. Two-sided *p* values less than 0.05 were considered significant.

## 3. Results

Between September and December 2020, 104 patients with confirmed SARS-CoV-2 infection were enrolled. Patients had a mean age of 68.72 ± 18.32 years and a high severity of illness. The Charlson Comorbidity Index was 4.42 ± 3.34, indicating a moderate–severe grade of comorbidity. Mean Simplified Acute Physiology Score (SAPS II) [37] was 33.66 ± 14.5. Mean Richmond Agitation-Sedation Scale [38] was −1.3 ± 2.1. Hypertension (63.46%) was the most common disease, followed by diabetes (23.12%) and chronic kidney disease (21.15%). Mean ICU length of stay (LOS) was 11.51 ± 7.51. A total of 48 patients (46.15%) died during hospitalization. Principal clinical characteristics of enrolled patients are summarized in Table 1 and Table 2.

Data are mean ± SD unless otherwise indicated. ICU-LOS (Intensive Care Unit Length of Stay), SAPS II (Simplified Acute Physiology Score), RASS (Richmond Agitation-Sedation Scale), and Healthy negative PCR controls (N = 53) are included and are age- and sex-matched with patients (33 males, mean age 56.12 ± 9.64). The non-survivor patients had C-reactive protein at admission and after 72 h, procalcitonin after 72 h, and white blood cells at admission higher than the survivors (*p* = 0.004, *p* = 0.001, *p* = 0.042, *p* = 0.027, respectively).

Based on the previous results of the first wave of COVID-19 infection, a comparison was first made with the IL-6, sIL-6R, and sgp130 levels in serum samples from control and patient groups.

As shown in Table 3, the mean values of IL-6 and sIL-6R were higher in COVID-19 patients than in the healthy control group.

IL-6 concentration in patients serum (265.51 ± 976.82 pg/mL) was significantly (*p* < 0.0001) higher (about 145 fold) than in the control group (1.92 ± 0.58 pg/mL).

In addition, sIL-6R in patients (39.71 ± 17.87 ng/mL) was significantly (*p* < 0.005) higher (about 1.3 fold) than the control group (30.01 ± 7.68 ng/mL).

Contrarily, the amount of sgp130 in serum patients (181.52 ± 67.29 ng/mL) was 59% with respect to the control group (324.31 ± 43.58 ng/mL) (*p* < 0.0001).

The levels of IL-6, sIL-6R and sgp130 in serum samples were then compared between the two patient groups with different outcomes: group 1 (survivors) and group 2 (non-survivors), as compared to the control group. 

In Table 4, parametric and nonparametric correlations between levels of IL-6, sIL-6R, and sgp130 with Charlson SAPSII and RASS scores are reported; no significative correlations were found.

As shown in Figure 1, the values of IL-6 were dramatically higher in non-survivors (291.90 ± 1421.01 pg/mL) than in survivors (72.91 ± 301.63 pg/mL) (*p* < 0.0001).

In the same way, as shown in Figure 2, sIL-6R in non-survivors (42.18 ± 20.42 ng/mL) was significantly (*p* < 0.05) higher than in the survivor group (37.74 ± 20.66 ng/mL).

In addition, Figure 3 shows that the amount of sgp130 in the nonsurvivors serum (172.52 ± 81.71 ng/mL) was significantly lower (*p* < 0.0001) than in that of the survivors (207.71 ± 88.93 ng/mL).

Evidently, each amount of IL-6, sIL-6R, and sgp130 obtained in the survivor and non-survivor groups was significantly modified with respect to the healthy control group.

Patients with five points of the Charlson Comorbidity Index (CCI) had statically significant higher levels of IL-6 at day 1, day 2, and day 3 and lower levels of sIL-6R at day 1 (Figure 4).

## 4. Discussion

SARS-CoV-2 binds to alveolar epithelial cells, activating the innate and adaptive immune systems and causing the release of a large number of cytokines, including IL-6. The results of our study demonstrate that, during the second wave of COVID-19 infection, patient IL-6 serum levels were significantly elevated in comparison with healthy, non-COVID patients, and increased IL-6 levels were significantly associated with adverse clinical outcomes. This suggests that the progression of SARS-CoV-2 infection may be the consequence of an excessive host immune response and autoimmune injury. Patients able to overcome the disease showed decreased IL-6 and sIL-6R levels compared to patients who died during hospitalization in ICU. At the same time, survivors showed higher sgp130 values (207.71 ± 88.93 ng/mL) compared to the non-survivors (172.52 ± 81.71 ng/mL).

Cytokine storm and immunitary imbalance are common in septic patients admitted to ICU [39], and our results confirm the hypothesis of the importance of cytokine storms in severe COVID-19 cases, which supports the role of agonistic and antagonistic molecules during the activation of the immune response against SARS-CoV-2 infection. This indicates that enhancements in serum concentrations of IL-6 and sIL-6R during infection lead to an increased agonistic trans-signaling mechanism. At the same time, lower sgp130 that interacts with the IL-6/sIL-6R-complex leads to a decreased block in IL-6 trans signaling, thereby confirming a reduced antagonistic role of sgp130 during the viral infection. 

After a decrease in detected cases in the summer of 2020, our ICU unit in Italy faced a second wave of COVID-19, which turned out to be less serious than the first, as indicated in the death/infection ratio. Our previously published results [40] showed cytokine determinations obtained during the first wave of SARS-CoV-2 infection, whereby we observed that serum levels of IL-6 (702.99 ± 1932.54 pg/mL) and sIL-6R (90.42 ± 74.29 ng/mL) in patients were significantly higher than in healthy control groups (1.81 ± 0.89 pg/mL and 29.91 ± 8.16 ng/mL, respectively). Contrarily, serum sgp130 levels were decreased in patients (217.92 ± 53.14 ng/mL) compared to the control group (305.24 ± 44.99 ng/mL).

Between the first and second waves, different treatments, including the use of high doses of corticosteroid, were administered to severe COVID-19 patients. However, during the second wave of infection, it is reasonable to assume that better and faster treatments with corticosteroids may have played a protective role in patients admitted to ICU, indicating a lower increase of IL-6 (265.51 ± 976.82 pg/mL) and sIL-6R (39.71 ± 17.87 ng/mL), despite a decrease in sgp130 levels (181.52 ± 67.29 ng/mL), with respect to the levels observed in the first COVID wave. The Charlson Comorbidity Index (CCI) was able to predict patient severity and mortality according to patient age, gender, and the presence of comorbidities such as diabetes, cardiovascular, cerebrovascular, and respiratory diseases, showing an exponential increase in the odds ratio of disease severity and mortality with each score point, whereas advanced age and multiple comorbidities were considered independent risk factors [41]. 

Our study found that, according to CCI, COVID-19 patients with several comorbidities had higher levels of IL-6 at day 1, day 2, and day 3 and lower levels of sIL-6R at day-1 only. Patients with multiple comorbidities were the most vulnerable in the COVID-19 patients, and the dis-regulation of IL-6 and its receptor may have worsened the prognosis. A patient’s CCI score is certainly a risk factor for mortality as indicated in the literature [42], as confirmed by our study results of OR 0.465; 95% CCI 0.281–0.648 *p* = 0.000.

In conclusion, the contemporary changes in IL-6, its receptors’ concentrations, and the consequent increased signal transduction, together with a high CCI score as a risk factor, are closely associated with mortality and unfavorable outcomes in COVID-19 infected patients. We can assume that these molecules, together with other pro-inflammatory molecules, contribute to the progression and severity of COVID-19 disease, whereas a Higher Charlson Comorbidity Index is associated with increased mortality and disease severity; yet, further investigations are required to evaluate the entire mechanism of action of a cytokine storm during COVID-19 infection. From a clinical point of view, as confirmed by our findings, the biomarkers included in this study may be useful tools for monitoring the anti-inflammatory activities of treatment options in COVID-19 patients. Further studies with a larger sample size may further correlate the contributions of these biomarkers with patient comorbidity.

## Figures and Tables

**Figure 1 pathogens-12-01264-f001:**
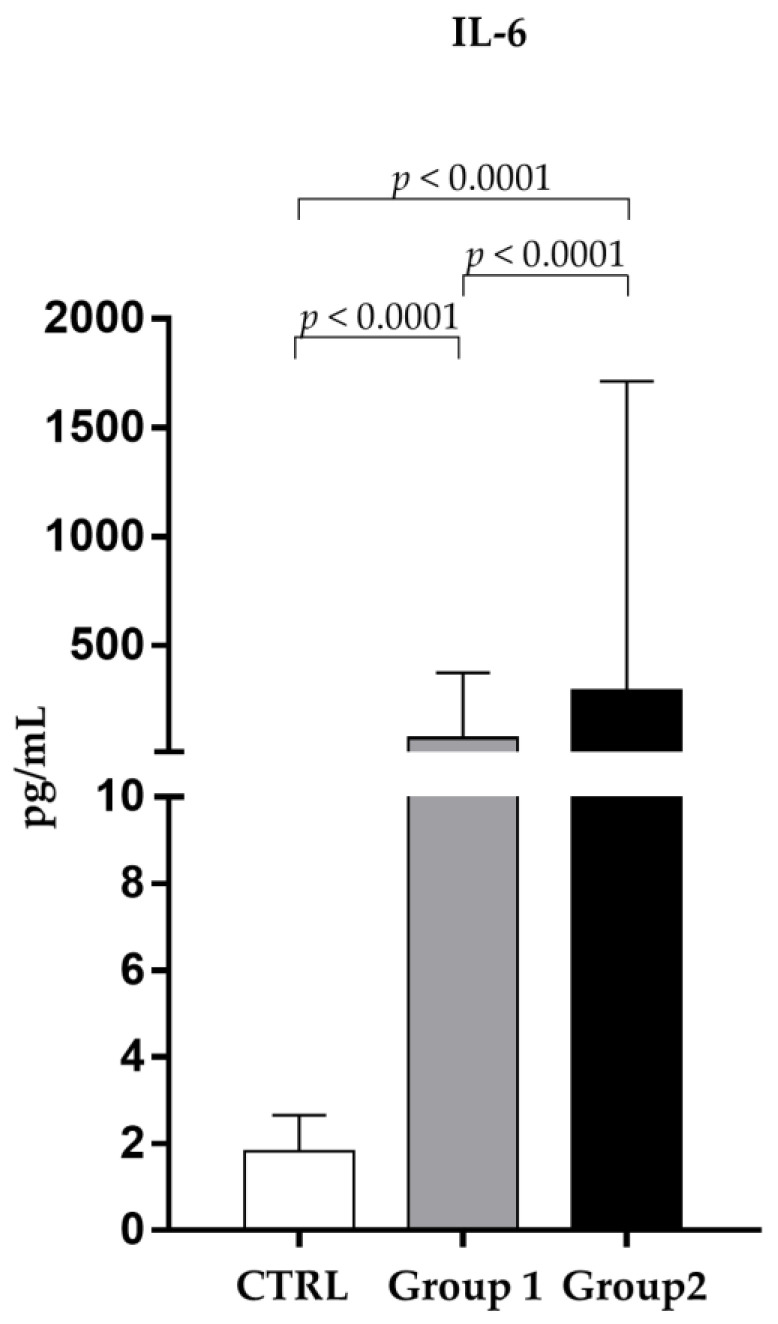
Analysis of IL-6 levels (pg/mL) in observed patients (mean ± S.D.) Group 1 (survivors): IL-6 levels (72.91 ± 301.63); Group 2 (non-survivors): IL-6 levels (291.90 ± 1421.01); CTRL (1.92 ± 0.58).

**Figure 2 pathogens-12-01264-f002:**
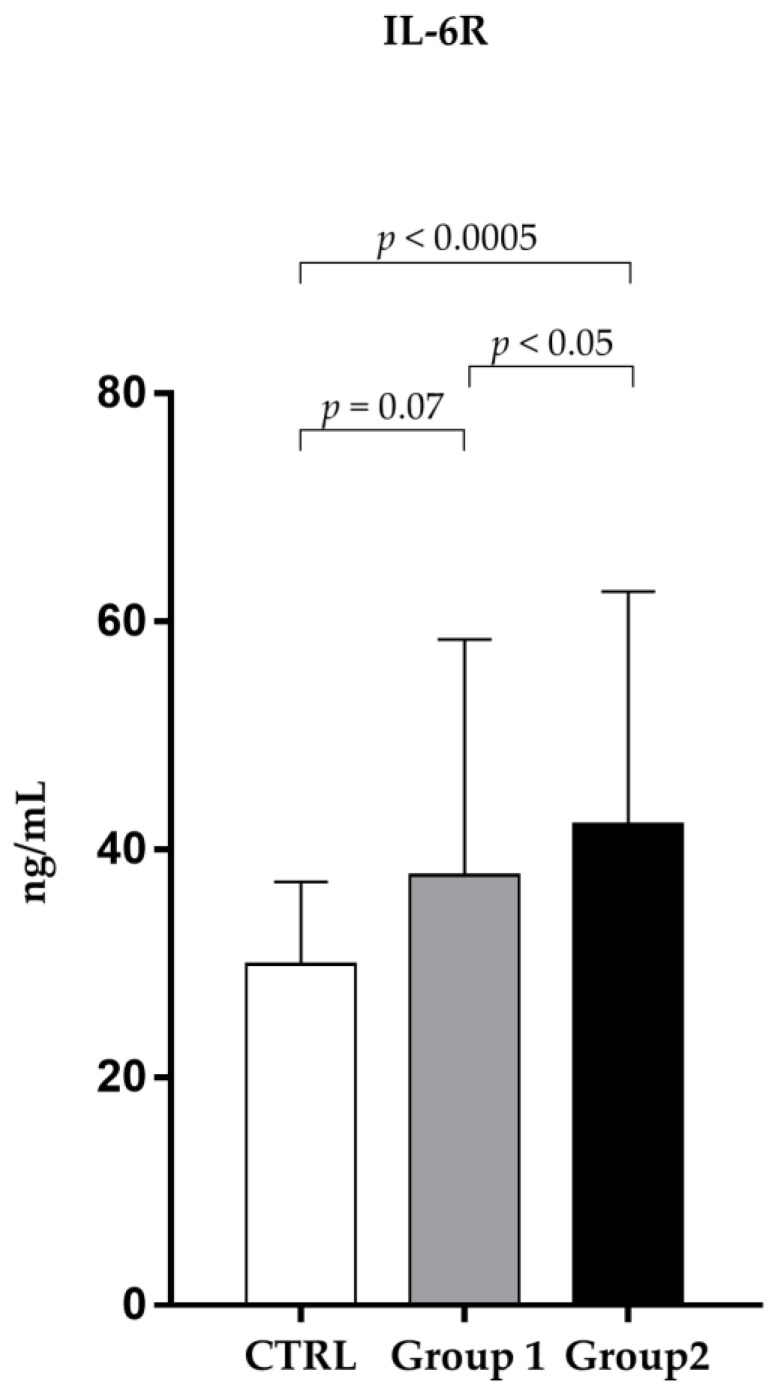
Analysis of sIL-6R levels (ng/mL) in observed patients (mean ±S.D.) Group 1 (survivors): sIL-6R levels 37.74 ± 20.66); Group 2 (non-survivors): sIL-6R levels (42.18 ± 20.42); CTRL (30.01 ± 7.68).

**Figure 3 pathogens-12-01264-f003:**
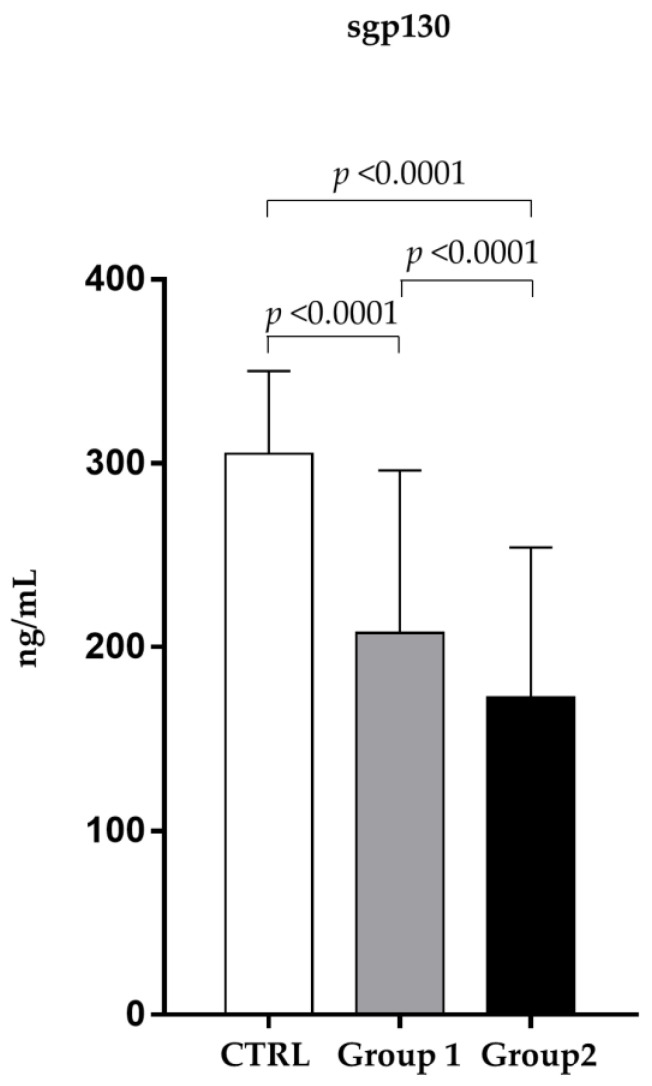
Analysis of sgp130 levels (ng/mL) in observed patients (mean ± S.D.) Group 1 (survivors): sgp 130 levels (207.71 ± 88.93); Group 2 (non-survivors): sgp 130 levels (172.52 ± 81.71); CTRL (324.31 ± 43.58).

**Figure 4 pathogens-12-01264-f004:**
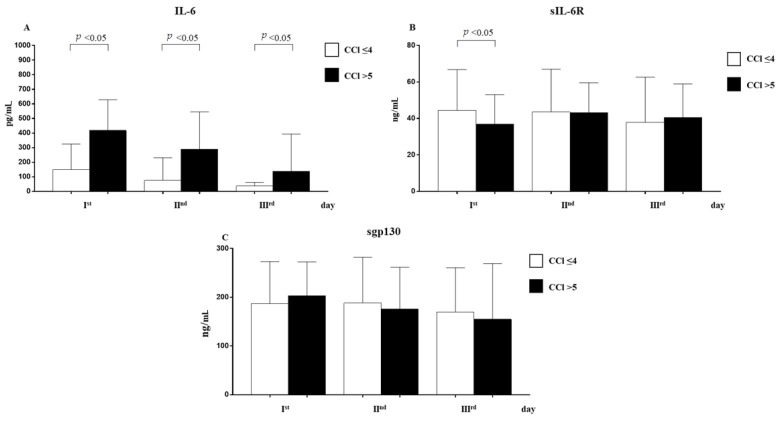
Analysis of IL-6 (**A**), sIL-6R (**B**), and sgp130 (**C**) levels in observed patients with Charlson Comorbidity Index ≤ 4 and >5 (mean ± S.D.).

**Table 1 pathogens-12-01264-t001:** Demographic and Clinical characteristics of enrolled patients.

Patients	N = 104
Age (years)	68.72 ± 18.32
ICU-LOS (days)	11.51 ± 7.51
Charlson comorbidity index (points)	4.42 ± 3.34
SAPS II (points)	33.66 ± 14.51
RASS (points)	−1.3 ± 2.1
Male (%)	68 (65.38%)
Diabetes (%)	24 (23.12%)
Hypertension (%)	66 (63.46%)
Chronic kidney disease (%)	22 (21.15%)
Death (%)	48 (46.15%)

**Table 2 pathogens-12-01264-t002:** Demographic and clinical data of surviving and non-surviving patients.

Patients		Survivors	Non-Survivors	*p*-Value
Age (years)		60.32 ± 15.12	72.51 ± 11.34	n.s.
Sex (%)	F	17 (18.7%)	19 (9.8%)	*p* = 0.543
	M	43 (42.2%)	25 (30.4%)	n.s.
Length of stay (days)		10.36 ± 6.21	11.42 ± 8.19	*p* = 0.299
Charlson comorbidity index (points)		2.36 ± 3.81	8.49 ± 2.93	n.s.
SAPS II (points)		24.52 ± 9.44	43.15 ± 17.51	n.s.
RASS (points)		0 ± 1	−3 ± 3	n.s.
Diabetes (%)		11 (10.9%)	13 (11.9%)	*p* = 0.202
Hypertension (%)		33 (34.7%)	22 (25.7%)	*p* = 0.612
Chronic kidney disease (%)		7 (6.9%)	15 (17.8%)	n.s.
C-reactive protein at admission (mg/L)		79.8 ± 81	125.9 ± 86.3	0.004
C-reactive protein after 72 h (mg/L)		59 ± 60.3	103.5 ± 72.6	0.001
Procalcitonin at admission (ng/mL)		3.26 ± 6.68	2.67 ± 6.97	0.796
Procalcitonin after 72 h (ng/mL)		0.22 ± 0.2	2.82 ± 1,24	0.042
White blood cell counts at admission (10^9^/L)		11.62 ± 6.74	15.74 ± 9.01	0.027
White blood cell counts after 72 h (10^9^/L)		11.82 ± 6.01	13.05 ± 6.26	0.299

n.s.: not significant.

**Table 3 pathogens-12-01264-t003:** IL-6, sIL-6R, and sgp130 levels in serum of healthy controls and patients (mean ± S.D.).

	Controls	Patients	*p*-Value
IL-6 (pg/mL)	1.92 ± 0.58	265.51 ± 976.82	*p* < 0.0001
sIL-6R (ng/mL)	30.01 ± 7.68	39.71 ± 17.87	*p* < 0.005
sgp130 (ng/mL)	324.31 ± 43.58	181.52 ± 67.29	*p* < 0.0001

**Table 4 pathogens-12-01264-t004:** Parametric and non-parametric correlations between SAPS II, Charlson index, and RASS with IL-6, sIL-6R, and sgp-130.

	Correlations	SAPS II	Charlson Index	RASS
IL-6	Person, *p*	−0.057, 0.568	−0.015, 0.882	0.068, 0.497
	Spearman, *p*	0.017, 0.87	0.026, 0.795	−0.051, 0.609
sIL-6R	Person, *p*	0.052, 0.61	0.139, 0.168	−0.155, 0.124
	Spearman, *p*	0.058, 0.567	0.106, 0.294	−0.141, 0.162
sgp-130	Person, *p*	−0.029, 0.773	−0.036, 0.722	−0.086, 0.392
	Spearman, *p*	−0.055, 0.588	−0.07, 0.491	−0.071, 0.485

## Data Availability

Data presented in this paper are available on request to the corresponding author.

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
