# Peer review of "Interleukin-6 and Its Soluble Receptor Complex in Intensive Care Unit COVID-19 Patients: An Analysis of Second Wave Patients"

_pathogens, 2023, doi:10.3390/pathogens12101264_

Round 1

Reviewer 1 Report (Previous Reviewer 1)

Many thanks to the authors for answering the questions raised. Yours results are as expected according to other publications

Minor editing of English language required

Author Response

Thank you for suggestions. The paper has now been revised by a mother-tongue English translator for a clearer and more concise text. 

Reviewer 2 Report (New Reviewer)

The manuscript under review is an interesting study on the role of IL6 and its receptor in the cytokine storm caused by COVID-19. However, I have a few major comments for the authors.

1) the introduction is too long. it provides a lot of useless information that can be took for granted and are not relevant for the study. pleae, make it shorter. 

2)in materials and methods, it is not explain when the study has been performed. in which time period?

3) in materials and methods, section "Patients", I think it should be explained when the blood samples occurred. it is only written "at five different times during hospitalization". which times?

4) I don't think the comparison between survivors and non survivors is the only useful information the authors could provide. for example, i think i would be more interesting if the authors had performed a comparison between the 104 severely ill patients and other patients that did not manifest severe covid-19. why the authors only concentrated on those 104 patients? in page 5/11, the authors introduced a "controls" group. this group has not been mentioned in the materials and methods. who are the patients of the controls group? could the authors provide more information about them? this is not clear from the manuscript. I think this is a huge problem of the manuscript.

 5) in the abstract, some sentences are repeated. please, correct.

The quality of English is quite poor, I think the manuscript needs a whole english revision. some sentences are too long, the meaning is sometimes diffucult to understand.

Author Response

1)The introduction is too long. it provides a lot of useless information that can be took for granted and are not relevant for the study. please, make it shorter.

Re: Thank you for suggestions. We have shortened the introduction for more concise reading, including only information relevant to our study.  

2) In materials and methods, it is not explained when the study has been performed. in which time period?

Re: The study was performed from September to December 2020 as already specified under the Patients section.  Page 3 lines 108-110.

3) In materials and methods, section "Patients", I think it should be explained when the blood samples occurred. it is only written "at five different times during hospitalization". which times?

RE: We have now modified this sentence specifying the blood sample intervals.  Blood samples for each patient were collected for 3 times every 48 hours from hospitalization. Page 3 lines 139-141.

4) I don't think the comparison between survivors and non survivors are the only useful information the authors could provide. for example, I think I would be more interesting if the authors had performed a comparison between the 104 severely ill patients and other patients that did not manifest severe covid-19. why the authors only concentrated on those 104 patients? in page 5/11, the authors introduced a "controls" group. this group has not been mentioned in the materials and methods. who are the patients of the controls group? could the authors provide more information about them? this is not clear from the manuscript. I think this is a huge problem of the manuscript.

Re: In this study we have considered only severely ill patients manifest severe Covid 19, admitted to our ICU (see Patients section).

In the “Materials and Methods” section we have provided more specific information about the healthy control subjects enrolled in the study (N=98; mean age 62.13±9.28). Page 3 line 137-139.

 5) In the abstract, some sentences are repeated. please, correct.

Re: We have modified and shortened the abstract accordingly.

Comments on the Quality of English Language

The quality of English is quite poor, I think the manuscript needs a whole English revision. some sentences are too long, the meaning is sometimes difficult to understand.

The manuscript has now been revised by an English language mother-tongue translator and has been made to read more clearly and concisely.

Reviewer 3 Report (New Reviewer)

This manuscript by Di Spigna et al explore the relationship of IL-6 and its soluble receptor complex (sIL-6R and sgp130) in critically ill Covid-19 patients with severe respiratory failure during the second wave outbreak. They found Covid-19 patients with several comorbidities including diabetes, hypertension and CKD had higher levels of IL-6 at day-1, day-2, day-3 and lower level of sIL-6R at day-1. The paper confirmed the presence of IL-6 during Covid-19 cytokine storm which is well published.

General comments:

1.     Introduction and Discussion, please include IL-6 modifiers, namely Tocilizumab, and to a lesser extent, Sarilumab as part of the Covid-19 therapy.

2.     Patients and methods, brief description of the Charlson’s scoring system, together with reference. The authors should provide detection limits of IL-6, sIL-6R and sgp130 in their assay systems.

3.     Table 1-3, and throughout the manuscript, “,” and “.”, ie, decimal point. Important units are missing, ie, ICU-length of stay (Days?).

4.     Table 2, please provide additional clinical information of both Survivors and Non-survivors including CRP, LDH, FBC, SARS-CoV RT/PCR cycle for viral count, and level of anti-Covid-19 IgG Spike and Nuclear antibody levels, presumably some of these patients would have been vaccinated/boosted. In additional, any information on abnormal CXR, ie, consolidation/opacities.

5.     Any correlation between levels of IL-6, sIL-6R and sgp130 with Charlson, SAPSII and RASS scores?

6.     OR results (line 253/254) – please consider moving under the Results section.

Minor comment.

Table 1, please replace “died” with “death”.

Please consider merging individual sentences into paragraphs.

Author Response

  1. Introduction and Discussion, please include IL-6 modifiers, namely Tocilizumab, and to a lesser extent, Sarilumab as part of the Covid-19 therapy.

      Re.  We have added in Introduction that different therapy, such as tocilizumab, sarilumab and steroids are now allowed to modulate the hyperinflammatory activation in COVID-19 infection. Page 2 lines 76-78.

  1. Patients and methods, brief description of the Charlson’s scoring system, together with reference. The authors should provide detection limits of IL-6, sIL-6R and sgp130 in their assay systems.

Re: We have added a description and references for the Charlson Comorbidity Index in the Patients section (lines 121-125) and included the detection limits of: IL-6 (0,01÷350,24 pg/ml), sIL-6R (3,01÷232,82 ng/ml), sgp130 (0,01÷2,40 ng/ml) in the Methods section (lines 142-143).

3) Table 1-3, and throughout the manuscript, “,” and “.”, ie, decimal point. Important units are missing, ie, ICU-length of stay (Days?).

Re: We have modified punctuation using the periods for decimals and added the ICU length of stay in the tables.

  1. Table 2, please provide additional clinical information of both Survivors and Non-survivors including CRP, LDH, FBC, SARS-CoV RT/PCR cycle for viral count, and level of anti-Covid-19 IgG Spike and Nuclear antibody levels, presumably some of these patients would have been vaccinated/boosted. In additional, any information on abnormal CXR, ie, consolidation/opacities.

Re: We have added more clinical information about the 2 groups in table 2. We did not have the RT/PCR cycle for viral count and level of anti-Covid-19 IgG Spike and Nuclear antibody levels at the time of the study (September -December 2020) as they were not routinely implemented at our institution at that time but were only available for a few patients and only after a specifically motivated request.

Regarding vaccinations, our study was performed between September and December 2020. At that time the COVID-19 vaccine was not available in Italy. The vaccination for COVID-19 was only made available for administration in January – February 2021.

  1. Any correlation between levels of IL-6, sIL-6R and sgp130 with Charlson, SAPSII and RASS scores?

Re: We did not find any parametric and nonparametric correlations between levels of IL-6, sIL-6R and sgp130 with Charlson, SAPSII and RASS scores, so we decided to not include them in the paper.

  1. OR results (line 253/254) – please consider moving under the Results section.

Re: We have moved this data to the Results section.

Minor comment.

Table 1, please replace “died” with “death”.

Re: We have corrected this table.

Comments on the Quality of English Language

Please consider merging individual sentences into paragraphs.

We have now merged sentences and paragraphs for a more streamlined manuscript format.

Round 2

Reviewer 1 Report (Previous Reviewer 1)

Many thanks to the authors for the corrections made, which have improved the clinical information and the discussion of the results.

Author Response

We have improved the clinical informations in table 2 and table 4 (highlighted in yellow).

Reviewer 2 Report (New Reviewer)

The authors adressed the suggestions, the manuscript has improved, 

The quality of english is better now. 

Author Response

Thank you for your comments and suggestions.

Reviewer 3 Report (New Reviewer)

We did not find any parametric and nonparametric correlations between levels of IL-6, sIL-6R and sgp130 with Charlson, SAPSII and RASS scores, so we decided to not include them in the paper.

Please include the above findings in either the Results or Discussion, as these are important information stating the lack of correlation between biomarkers and clinical scores.

Clinical information of both Survivors and Non-survivors including CRP, LDH, FBC are missing in Table 2 as these are important information.

Finally, it should clearly state in the manuscript all patients are Covid-19 vaccine naive.

The current manuscript will benefit from further moderate English editing.

Author Response

For parametric and nonparametric correlations between biomarkers and clinical scores see lines 243-245 and table 4 (highlighted in yellow). 

For the clinical informations required (CRP, LDH and FBC) see table 2 (highlighted in yellow).

For Covid-19 vaccine see "Patients" lines 144-146 (highlighted in yellow).

Round 3

Reviewer 3 Report (New Reviewer)

Many thanks for including the additional data in Table 3 & Table 4. Table 3, what is the difference between the 2 WBCs at admission (last two rows highlighted in yellow, ie, 72hrs post admission?

Please provide a brief description of Table 3 under the results section, ie, CRP, procalcitonin & WBC are known markers for COVID-19 clinical outcome together with appropriate references.

Minor comment: Table 3 formatting

Author Response

Many thanks for including the additional data in Table 3 & Table 4. Table 3, what is the difference between the 2 WBCs at admission (last two rows highlighted in yellow, ie, 72hrs post admission?

Re: our apologies, the second row refers to WBC after 72 from admission. We correct the lines.

Please provide a brief description of Table 3 under the results section, ie, CRP, procalcitonin & WBC are known markers for COVID-19 clinical outcome together with appropriate references.

Re: we added those data in the results (highlighted in yellow). 

Minor comment: Table 3 formatting

Re: we corrected the table 3 that now is table 2.

Round 4

Reviewer 3 Report (New Reviewer)

No further comments

This manuscript is a resubmission of an earlier submission. The following is a list of the peer review reports and author responses from that submission.

Round 1

Reviewer 1 Report

You have measured IL-6 and its soluble receptor complex (sIL-6R and sgp130) in 104 critically ill COVID-19 patients with severe respiratory failure during the second wave. You state that the current data is like that obtained in a previous publication by your group (Clinical and Medical Investigations 2020). However, in this work you do not make a detailed comparison with the data from the previous publication. Neither do you analyze data from the third pandemic wave. Therefore, the phrase that appears in the title of the article does not seem justified. "a comparative analysis of different pandemic waves”. Nor can it state in the last sentence of the introduction: "to evaluate the possible differences in respect to the first wave”.  

2. The clinical characteristics of the survivors and non-survivors patients should be specified, since only the data of the patients as a whole are given. It would also be useful to detail the characteristics of the control group, for example BMI, renal and hepatic function.  

3. The results obtained are like those found in the first pandemic wave. But why would there be a different response in the second wave?. It is well known that a hyperimmune response is common in patients admitted to the ICU, therefore it should be explained why you expected to find differences.

4. In the hyperimmune response ("cytokine storm") triggered by SARS-COV-2, a cascade of pro- and counter-inflammatory cytokines is activated. It is well-known that several cytokines are closely associated with mortality and unfavorable outcome in patients infected with COVID-19. Why this study has only focused on the IL-6 response and in the classical signal transduction and trans-signal transduction of IL-6?.

5. The sentence in the Conclusion: "these molecules could be considered as promising markers for disease progression, ill severity and risk of mortality" does not seem specifically derived from the data of this study since indeed other studies have demonstrated the usefulness of these molecules as prognostic markers.

Moderate editing of English language is required

Author Response

Thank you to review the our manuscript. 

Reviewer 2 Report

Dear Editor,

Thank you for giving me the opportunity to review the manuscript titled "Interleukin-6 and its soluble receptor complex in COVID-19 patients in the Intensive Care Unit: a comparative analysis of different pandemic waves." The manuscript presents intriguing findings on cytokines in severely ill patients during the second wave at an Italian hospital and explores their potential as markers for disease progression. The manuscript is engaging and well-organized. However, I have some concerns, and the following comments address these issues:

Abstract, line 28: The statement "since there is no specific treatment for COVID-19" should be rephrased. Currently, there are antivirals available for COVID-19 treatment. However, it is possible that the authors meant to refer to the lack of specific treatments for severe cases in the later stages of the disease.

Introduction, line 100: The objective of this study was to analyze IL-6 and its soluble receptor complex in critically ill COVID-19 patients with ARDS during the second wave of COVID-19 infection. The goal was to assess potential differences compared to the first wave and propose a plausible mechanism for disease progression associated with various comorbidity factors. However, in the Results section, only the samples collected from September to December 2020 were presented. Therefore, the title, objective, and methods should be harmonized with the results to ensure consistency.

Discussion: While it is appropriate to compare the findings of this study with previous publications, the aim of this study should reflect the research conducted during the study period.

Finally, it is important to address the advancements/limitations in treatment options for these patients adequately in the Conclusion and the use of the study findings.

I hope these suggestions will help in improving the manuscript.

Best regards!

Dear Editor,

as usual, I recommend that the authors review the quality of the English language in their work, specifically for minor editing corrections.

Author Response

Thank you to review the our manuscript.

Reviewer 3 Report

Dear Authors of manuscript entitled "Interleukin-6 and its soluble receptor complex in COVID-19 patients in Intensive Care Unit: a comparative analysis of different pandemic waves”.

Thank for your idea to find some markers related disease morbidity.

However, I wish that you would correlate your results with more immunological, clinical and biological data to draw the story of IL-6 contribution to disease pathogensis and comorbidity.

Relying on IL-6, sIL6R and gp130 is not sufficient to get to some conclusions.

I deeply recommend to improve the quality of work to get to some new findings.

Best wishes.

/

Author Response

Thank you to review the our manuscript.
